# The Need to Strengthen the Role of the Pharmacist in Sri Lanka: Perspectives

**DOI:** 10.3390/pharmacy7020054

**Published:** 2019-06-05

**Authors:** M.H.F. Sakeena, Alexandra A. Bennett, Andrew J. McLachlan

**Affiliations:** 1Department of Pharmacy, Faculty of Allied Health Sciences, University of Peradeniya, Peradeniya KY 20400, Sri Lanka; 2Sydney Pharmacy School, The University of Sydney, New South Wales 2006, Australia; andrew.mclachlan@sydney.edu.au; 3NSW Therapeutic Advisory Group, New South Wales 2010, Australia; sasha.bennett@svha.org.au

**Keywords:** pharmacist, pharmacy practice, pharmacy education, community pharmacy, hospital pharmacy, Sri Lanka, developing country

## Abstract

The role of the pharmacist in healthcare has evolved greatly over the last half-century, from dispensing to providing direct patient-oriented activities not associated with dispensing. However, pharmacist-led healthcare services in Sri Lanka must undergo reform to fully take advantage of their expertise and training in medicine management and related outcomes in Sri Lankan patients. As befits a profession’s role development and value, professional and educational standards for pharmacists need ongoing development and growth. Currently, university curricula and continuing professional education in Sri Lanka require further development and optimisation to provide the theoretical and practical knowledge and skills regarding quality use of medicines and patient-oriented care. Furthermore, pharmacists’ roles in Sri Lankan hospital and community pharmacist settings need to be recognised and should include the pharmacist as an integral part of the multidisciplinary healthcare team in Sri Lanka. Studies from developed countries and some developing countries have demonstrated that expanded pharmacists’ roles have had a significant positive cost-effective impact on the population’s health. Therefore, the availability of qualified Sri Lankan pharmacists trained to deliver expanded professional services accompanied by greater pharmacist integration into healthcare delivery is crucially important to ensure quality use of medicines within the Sri Lankan healthcare system and optimise the medication-related needs of Sri Lankans.

## 1. Introduction

Pharmacy is a major health discipline that contributes to the development of a country [1]. Pharmacists play a vital role in pharmaceutical manufacturing, marketing and regulation of medicines [2], providing pharmaceutical care services [3], and disseminating their knowledge through academia and research [2,3]. Furthermore, pharmacists are important members of the multidisciplinary healthcare team caring for patients and playing a major role in ensuring the judicious use of medicines [4,5]. Expanded clinical pharmacists’ roles contribute to cost-effective therapeutic outcomes [6]. These roles are well-recognised and considered integral parts of many countries’ healthcare systems [6,7,8], including those of some developing countries [9]. However, at present, pharmacists are not yet an integral part of the multidisciplinary healthcare team in Sri Lanka, and their role is currently limited to drug dispensing, compounding and providing limited medicine information to patients [10]. 

In order to achieve the ultimate goal of improving the healthcare services of a nation, the pharmacy profession should be committed to continuously improving its knowledge and quality of practice. The aim of this chapter is to highlight the need to strengthen the roles of pharmacists in Sri Lanka. Further, this chapter will describe the importance of pharmaceutical care services, the current scenario of pharmacists’ roles in hospital and community pharmacies in Sri Lanka and how further augmentation of pharmacy education can produce competent pharmacists able to deliver enhanced pharmaceutical care across all Sri Lankan healthcare settings and ultimately optimised patient self-management of medicines. 

## 2. Pharmacists’ Roles in Sri Lanka: The Current Scenario of Pharmaceutical Care Service Delivery 

The role of the pharmacist around the world has evolved greatly over the last decades, expanding from dispensing activities to enhanced direct patient-care tasks, administrative activities and public health functions. Pharmaceutical care aims to provide responsible and careful medication therapy to achieve optimal medication outcomes and increase patient satisfaction in an integrated health system that ultimately improves the quality of a patient’s life [11]. However, in many developing countries, qualified pharmacists have historically played a more pharmaceutical industry-oriented role rather than a patient-oriented perspective [2]. Similarly, in Sri Lanka, pharmacists are well recognised and well remunerated for their activities in the pharmaceutical industry. Furthermore, pharmacy services in the hospital and community in Sri Lanka tend to be limited to medicines procurement, medication supply and dispensing, inventory control and storage [12]. Compared to many countries, the pharmacists’ role in Sri Lanka is under-recognised and underutilised, and pharmaceutical care is yet to be recognised as a field of pharmacy practice [13]. This is despite evidence that improved pharmaceutical care services can be delivered in the Sri Lankan healthcare setting [14]. 

Official Sri Lankan reports have made recommendations for the improvement of pharmaceutical care services in both hospital pharmacy and community pharmacy settings, albeit with little fruition [15]. Currently, dispensing of prescriptions is the primary duty of hospital and community pharmacists in Sri Lanka. Compounding is also undertaken in most hospitals and a few community pharmacies. Registration to practise as a pharmacist is obtained from Ceylon Medical Council after awarding of a certificate of proficiency in pharmacy, a certificate of efficiency in pharmacy a diploma in pharmacy or a degree in pharmacy from a recognised institution or university in Sri Lanka or abroad. 

### 2.1. Hospital Pharmacy

Pharmaceutical care services in the hospital setting promote appropriate medicine use and cost-effective prescribing and enhance medication safety [16]. Enhanced hospital pharmacy activities have helped to optimise treatment, promote rational prescribing and reduce inappropriate medicines use and preventable medication related harms and improve patient outcomes [17,18]. A variety of pharmacist-led interventions to improve medicine use in the hospitals have been established [19]. These programs usually involve multidisciplinary teamwork and often incorporate pharmacists with postgraduate training to facilitate appropriate medicines use in hospitalised patients [20]. Many countries have achieved success with the implementation of clinical pharmacy programs [21], resulting in improved clinical and economic outcomes [6]. However, clinical pharmacy services are not widely available in many developing countries [22]. The reasons for a lack of clinical pharmacist-led activities in hospitals in these countries are complex but may reflect the emerging nature of the pharmacy profession in these countries, professional isolation and a lack of recognition of the role of clinical pharmacy within the healthcare system [23].

In recent years, developing countries such as Malaysia [24], Taiwan [25], Brazil [26], Kuwait [27], Sudan [28], Nigeria [29] and Ethiopia [30] have realised the importance of pharmaceutical care service delivery in hospitals and implemented clinical pharmacy programs. Empirical studies reveal that implementation of clinical pharmacy services reduce the length of hospital stays and produce significant cost savings, both medication costs and total inpatient expenditures [22,31]. These developments have the potential to lead to improved patient quality of life, better use of healthcare resources and a reduced rate of mortality [31]. Importantly, clinical pharmacy programs are emerging in some South Asian countries. For example, clinical pharmacy programs have been reported in a South Indian hospital [32] and two hospitals in Pakistan (Shaukat Khanum and Agha Khan hospitals) [33]. Furthermore, the International Pharmaceutical Federation (FIP) Basel Statements [34] cover all areas of medication management in a hospital setting, including procurement, preparation and delivery, as well as prescribing, administration and the monitoring of patient outcomes and human resources. This FIP guideline represents an important opportunity to identify gaps in Sri Lankan hospital pharmacy practice according to international standards and to identify priority areas for implementation and workforce development. 

There are only a few studies and reports in the literature discussing Sri Lankan hospital pharmacy and pharmaceutical care service delivery. Currently, only a small number of pharmacists are employed in Sri Lankan hospitals [13], and pharmacists are typically not a part of the multidisciplinary healthcare team [14]. In the Sri Lankan hospital setting, the pharmacy department is usually divided into “outpatient” and “inpatient” pharmacies. The outpatient pharmacy dispenses medicines to clinic patients and patients from the outpatient department, while the inpatient pharmacy distributes medicines to the hospital wards and inpatients. As the hospital adopts a ward-stock method of medication distribution, pharmacists in the inpatient pharmacy do not come into direct contact with patients, prescribers and prescriptions. The pharmacist’s activities in hospitals are limited to drug procurement, distribution, inventory control and drug dispensing for both types of pharmacies, with additional activities such as limited medicines information provision and compounding for patients [14,35]. A ward-based clinical pharmacy services does not currently exist in Sri Lankan public hospitals [14]. However, a recent report suggests that there are significant opportunities for pharmacists to provide pharmaceutical care services in Sri Lankan hospitals [10]. Furthermore, this study found positive attitude from doctors regarding the inclusion of a ward-based clinical pharmacist to the healthcare team in Sri Lanka.

A high prevalence of preventable medication errors (35.5%) related to incorrect dosing regimens and inappropriate documentation for prescribed or administered medications inpatient records has been reported in Sri Lankan hospitals [36]. This represents a major safety concern for inpatients’ treatment in hospital [36]. Further studies from Sri Lanka have concluded that error-prone abbreviations and inappropriate, unapproved abbreviations may be frequently used in prescription writing [37], as well as incomplete outpatient handwritten prescription writing [38], leading to medication errors. Another study from Sri Lanka found that patients who lacked their medication record or possessed an incomplete record posed a significant barrier to appropriate medication use among the elderly patients [39]. Thus, patient safety and the health and quality of life of Sri Lankan hospital patients currently appear compromised. Implementation of pharmaceutical care or clinical pharmacy services is a highly feasible solution to overcome these health challenges and, as has been previously shown, may lead to reduced preventable harms, improved quality use of medicines in Sri Lankan hospitals and reduced total healthcare costs.

Using pharmacists to provide pharmaceutical care services in hospitals and recognising them as part of the healthcare team may improve access to comprehensive medicines information for patients and clinicians and enhances the ability for pharmacists to develop close collaborative relationships with prescribers and other healthcare professionals [40]. Further, pharmacists can assist in optimising medication management, direct patient care, public health initiatives, population management activities and the provision of expert drug information and education for other primary healthcare team members [41,42]. Counselling of patients by pharmacists about appropriate medicine use is an important role for pharmacists [43] and can help to improve medication adherence [44]. Pharmacists with good communication and change behaviour skills can also improve adherence to clinical therapeutic guidelines. These positive outcomes would be achievable in Sri Lanka by strengthening the role of pharmacists and pharmaceutical care services provision. The international evidence supporting the role of the hospital pharmacists in improving healthcare provides a strong rationale for calling on government authorities to take action to introduce and implement enhanced pharmacy services in hospitals nationwide.

### 2.2. Community Pharmacy

Community pharmacists play a vital role in healthcare provision [45]. Community pharmacists are often the first contact person for patients seeking advice on common health complaints [46]. They also provide advice on the self-management of minor ailments [47] or provide triage and referral to other health professionals. Furthermore, pharmacists may be the last contact person before the patient commences any medication for treatment, so they have a role in checking for drug allergies, reinforcing the messages regarding appropriate use of medicines, ensuring adequate monitoring, providing guidance on adverse drug effects as well as drug interactions, and improving patients’ medication adherence [48]. 

In developed countries, community pharmacy services are well recognised and utilised [49], and the community pharmacists’ role is integrated into the healthcare system [8,46]. For example, in the USA, community pharmacists are a readily accessible primary healthcare provider [7]. In Australia, community pharmacists are considered educators regarding the quality use of medicines for the public [46], and in Canada, community pharmacists play a proactive role as providers of primary healthcare services [49]. Further, to promote self-care and to improve the management of chronic health conditions, the United Kingdom has introduced the New Medicine Service (NMS) to be delivered by community pharmacists [50]. 

By contrast, community pharmacy services offered in developing countries are typically confined to traditional practices of dispensing with a business-oriented approach [2] and a limited emphasis on patient health and welfare [3]. Community pharmacists are also known as drug retailers [51,52]. Community pharmacists have a definite role in improving the quality use of medicines and the quality of life of their customer; however, it has been observed that community pharmacies in most low-income settings are often underutilised, and the standard of care and services tend to be variable and often suboptimal [53]. As a consequence, inappropriate dispensing procedures such as dispensing prescription-only medicines without an authentic prescription occur in most of the pharmacies in developing countries [54]. 

In Sri Lanka, privately-owned community pharmacies are the main source of medicines, and dispensing is mostly undertaken by pharmacists and drug retailers [55,56]. Currently, Sri Lankan community pharmacists are involved in procurement of medication, extemporaneous compounding, dispensing of prescriptions and selling of medicines [57]. Pharmacists are not commonly involved in providing pharmaceutical care services. Patients with chronic diseases often visit the pharmacies for their prescription refills; however, their disease conditions are not followed up or monitored by the community pharmacist, in a similar manner to many other developing countries [58]. Adverse drug reaction (ADR) reporting by pharmacists is now well established in many countries [59]; however, pharmacists are not involved in ADR reporting in Sri Lanka, despite their high prevalence [60]. Inappropriate labelling and insufficient information for patients during the dispensing of medications at community pharmacies [57] has been an ongoing challenge for many years.

The absence of a qualified and trained pharmacist in the community pharmacy setting is the mains reason for many of the observed inappropriate dispensing practices in Sri Lankan community pharmacies. Remuneration and recognition are two concerns for qualified pharmacists to seek work opportunities elsewhere [61]. Guidelines for community pharmacy practice issued by the National Medicines Regulation Authority (NMRA), Sri Lanka [62] stipulates that community pharmacies should have legally qualified pharmacy professionals at all times during opening hours. However, this rule is not strictly enforced; most pharmacies dispense medications without the presence of a qualified pharmaceutical professional [63]. A recent study conducted in Sri Lanka revealed that antibiotics can be purchased without prescriptions in different districts of Sri Lanka [64]. Moreover, this study found that the presence of a qualified pharmacist was associated with reduced nonprescription dispensing practices of antibiotics by Sri Lankan community pharmacies. Hence, enforcement of appropriate staffing in licensed pharmacists during business hours should be undertaken in order to ensure the high quality of medications and services. All pharmacy personnel should undergo formal training in dispensing of medicines, participate in continued professional development program and adhere to legal obligations related to the supply of medicines. 

There is a considerable need for healthcare services provided by community pharmacies in Sri Lanka and an urgency to undergo reforms to meet the changing needs of medicines users and align with international standards of practice. Apparently, most of the medication use and supply occurs in community-based settings [65], so it is important to understand pharmacists’ current role in patient care, and their perception of the quality of patient care services available in real-world community-based pharmacies. A recent systematic review highlighted that community pharmacist-led interventions have been shown to contribute to improved adherence and better disease control [66]. 

Furthermore, community pharmacists can play a key role in optimising medicines use to help achieve the quality use of medicines in the community. Importantly, pharmacists’ contributions to healthcare need to be recognised. There is an urgent need to strengthen community pharmacists’ role in Sri Lanka. To achieve this, policy changes in the healthcare system for pharmacists should be considered for adoption in Sri Lanka. Remuneration and recognition for the extended roles that a pharmacist can play (beyond supplying medicines) are critical in order that qualified pharmacists achieve their full scope of practice in community pharmacies. 

Figure 1 [45,46,67,68,69,70,71,72] represents the expanded pharmacists’ role delivered through different pharmaceutical care services to provide optimum healthcare outcomes. Th diagram shows that pharmacists’ role has enhanced to provide services in different sectors of healthcare delivery.

## 3. Pharmacy Education

Pharmacists with the appropriate education and training can assist both healthcare professionals and patients in using medications wisely [73,74,75]. Furthermore, comprehensive and relevant education and training on the quality use of medicines is essential for pharmacists to take a leading role in changing behaviours around medicines consumption and their appropriate use in the community [76].

### 3.1. From Science to Patient Care

Pharmacy undergraduate education in Sri Lanka is a predominantly science-based university course. If pharmacy education is to keep pace with current and future changes in practice, then pharmacy training must become more patient-focused, embracing elements of clinical and therapeutics training. There is a necessity to provide knowledge and skills on pharmaceutical care through pharmacy curriculum at undergraduate level [77]. A primary focus of pharmacy teaching should be on patient centred-care, training future pharmacists to lead quality use of medicines initiatives and patient-oriented pharmaceutical care services. Problem-based learning methodologies [78] can assist students in developing skills in clinical reasoning, patient safety and clinical accountability, ensuring that shared decision making, documentation and reporting are implemented in practice. To excel in pharmacy practice, pharmacy students should have greater exposure to first-hand clinical practice [79]. Pharmacy students should be able to practise and work within multidisciplinary environments [80]. Therefore, undergraduate pharmacy education in Sri Lanka should include the core subject of clinical pharmacy practice and instil the attributes of life-long learning to ensure pharmacists in practice maintain currency of their therapeutics skills and knowledge.

### 3.2. Postgraduate Training and Future Leaders

The next imperative is to provide postgraduate (PG) level training for pharmacists in Sri Lanka. These PG courses can be introduced as M. Pharm in Clinical Pharmacy or M. Pharm in Hospital Pharmacy or PhD in Pharmacy. Postgraduate courses focusing on the management of diseases and the development of quality use of medicines in Sri Lankan universities could be an additional strategy. Currently, pharmacy graduates undertake PG studies in allied fields, such as chemistry, since graduates do not have access to research or PG studies in the field of pharmacy or pharmacy practice. There are currently no academic institutions in Sri Lanka that offer postgraduate degrees in the pharmacy discipline, and this has meant Sri Lankan qualified pharmacists who wish to further their postgraduate pharmacy career need to undertake postgraduate studies abroad. To date, relatively few pharmacists have obtained Masters or PhD qualifications in pharmacy. This hampered development of pharmacy leaders with a wider range of competencies, knowledge and skills necessary for the development of future healthcare services in Sri Lanka presents pharmacy graduates with a significant barrier to their future academic and professional development and is a major challenge for the development of Sri Lankan pharmacy practice. It is essential that research and PG programmes in pharmacy-related fields are introduced. These future initiatives will not only provide benefits for the graduate pharmacists but will also have an important impact on the future development of pharmacy leaders and the pharmacy profession in Sri Lanka and optimise the appropriate, safe and effective use of medicines by Sri Lankans.

### 3.3. Pharmacy Curriculum

It is also equally important to analyse the contents of pharmacy curricula of undergraduate and postgraduate pharmacy programmes in other countries, which excel in pharmacy practice, such as Australia [81]. Australia has a number of requirements to ensure that educational standards are attained by all universities delivering courses for pharmacist qualification. These include mandated components in the university pharmacy curricula, an accreditation requirement for university courses every 5 years, and demonstration of professional competencies for registration as a pharmacist, overseen by a national pharmacy council [82]. These strategies should be considered by Sri Lankan academics and policy makers to raise the quality of pharmacy education and the performance of the pharmacy profession in Sri Lanka.

### 3.4. Continuing Professional Development

Additionally, continuous professional development courses and programs for practising pharmacists are another important strategy requiring establishment in Sri Lanka. Continuous professional education, using workshops, seminars and online courses is another important means to improve appropriate medication use and dispensing practices in Sri Lanka. These educational strategies could have a direct influence on patient-centred practices of pharmacists in Sri Lanka. If pharmacists are to consider providing enhanced pharmaceutical care services in future, then using the teaching and training of pharmacy practice at undergraduate level as well as maintaining the currency of their skills and knowledge through continuous professional courses will help to strengthen the role of pharmacists in Sri Lanka.

## 4. Conclusions

In Sri Lanka, pharmacists are currently underutilised, and their potential roles in patient-care can be viewed as a missed opportunity for optimising the health of the nation’s population. There is a greater need for Sri Lankan pharmacists to extend their services to offer pharmaceutical care. This could lead to the identification and resolution of drug-related problems, improved medication adherence, improved medication safety and enhanced patient care. To effectively and continually offer these services, national policies will need to be developed and enacted to provide some remuneration and recognition for the service. Several studies have demonstrated that the expanded role of pharmacists delivering cost-effective optimal patient care has a significant impact on the healthcare system of a nation. Therefore, the pharmacist’s role should be strengthened to deliver pharmaceutical care services in Sri Lanka. 

## Figures and Tables

**Figure 1 pharmacy-07-00054-f001:**
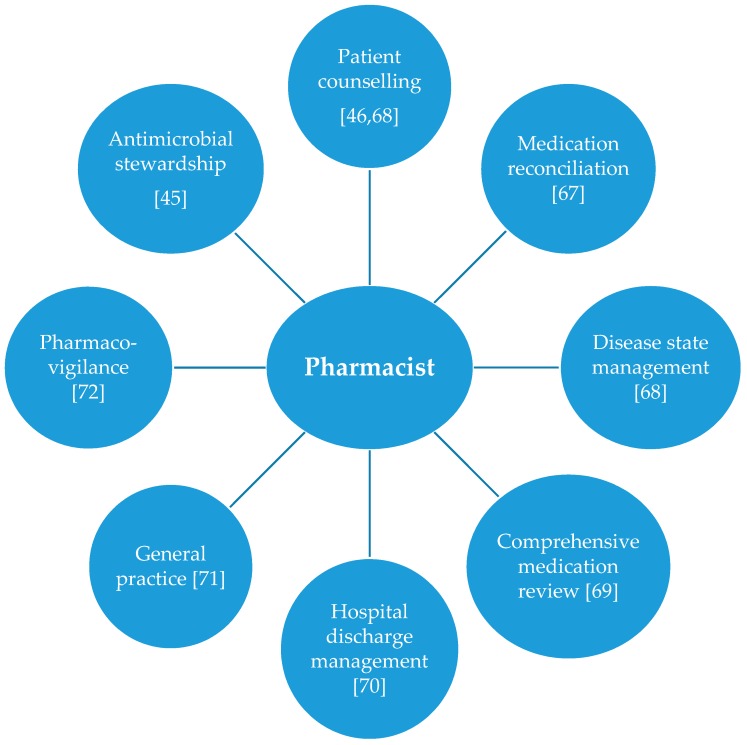
Expanded pharmacists’ role delivered through different pharmaceutical care services to provide optimum healthcare outcomes.

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
