# Peer review of "The Need to Strengthen the Role of the Pharmacist in Sri Lanka: Perspectives"

_pharmacy, 2019, doi:10.3390/pharmacy7020054_

Round 1
Reviewer 1 Report
This was an interesting perspective article to read which was well written and which made a coherent argument for expanded pharmacist roles and improved training in Sri Lanka. I have some comments below that the authors should consider to help clarify details and improve the manuscript.
Generally speaking, the authors could provide more specific examples of pharmacists impact from studies they cite when justifying the need for enhanced roles or education. This includes naming countries of origin, year, and main outcome results. This applies to all sections of the manuscript
Throughout the manuscript the authors use language that is too strong when supporting their case - language of certainty is used when in research it is more appropriate to be more speculative. I have provided many examples below but there are more in the paper that require addressing so careful review is required.
88 - in these countries are complex but may reflect
123- remove ‘many’
131 - may lead
134- may improve access
151- may be the last contact
209 - should be considered for adoption
124- may be frequently used
270 - could have a direct
273 - will help strengthen
279- could lead to
Section 2.2 should include supporting evidence that the population has outstanding health needs or is at risk from health concerns that the pharmacist could address in the community with an expanded role. In section 2.1 is there evidence that staff lack of knowledge regarding medicines is contributing to error that the pharmacist could address?
There were instances where references cited did not appear to directly support statements made in the text where they were used or were outdated and the authors should please update or cite the original studies. Examples include references 22 (line 87), 23 (90), 31 (95), 53 (170)
There are also additional systematic review studies of the pharmacists role that may be cited. E.g.
https://www.ncbi.nlm.nih.gov/pubmed/29723934
https://www.ncbi.nlm.nih.gov/pubmed/26149372
https://www.cochranelibrary.com/cdsr/doi/10.1002/14651858.CD013102/full
It is not clear where in the text figure 1 is cited - please can the authors make this clearer and also clarify what ‘General Practice’ means
The educational section is sparsely referenced in comparison to earlier sections and needs greater support with literature when advocating educational teaching methods and new PG courses - what is the impact on pharmacist learning outcomes, competency and/or care? What do regulators in other countries recommend? The Am J Pharm Education is a good source of literature in this area.
In the educational section, can the authors justify why they mention only M.Pharm or PHD when other qualifications are available. For example in the UK pharmacist can undertake prescribing qualifications.
The educational section undergraduate courses and experiential learning should be included in more detail with more supporting evidence of the positive impact of making such changes. In 3.4 please clarify if CPD is currently required to be completed or not by pharmacists
Please use the word ‘medication’ consistently in the text in place of ‘drug’ to improve clarity.
Further more minor comments and edits are below
Line 14 - what does significant mean? Clarify in the text please
Line 21 - include the pharmacist
285-6: last sentence not needed?
122- reference 36 please provide error rate
126- patients who lacked their medication record or possessed an incomplete record
Author Response
Reponses to reviewers’ comments
Ms. Ref. No.: Pharmacy 489555
Title:
The need to strengthen the role of the pharmacist in Sri Lanka: perspectives.
Pharmacy
REVIEWER 1:
This was an interesting perspective article to read which was well written and which made a coherent argument for expanded pharmacist roles and improved training in Sri Lanka. I have some comments below that the authors should consider to help clarify details and improve the manuscript.
Generally speaking, the authors could provide more specific examples of pharmacists impact from studies they cite when justifying the need for enhanced roles or education. This includes naming countries of origin, year, and main outcome results. This applies to all sections of the manuscript
Comment 1:
Throughout the manuscript the authors use language that is too strong when supporting their case - language of certainty is used when in research it is more appropriate to be more speculative. I have provided many examples below but there are more in the paper that require addressing so careful review is required.
88 - in these countries are complex but may reflect
123- remove ‘many’
131 - may lead
134- may improve access
151- may be the last contact
209 - should be considered for adoption
124- may be frequently used
270 - could have a direct
273 - will help strengthen
279- could lead to
Response
We have now revised the above suggestions in the revised manuscript for clarity and accuracy. This is presented in the track changed version of the manuscript.
Comment 2:
Section 2.2 should include supporting evidence that the population has outstanding health needs or is at risk from health concerns that the pharmacist could address in the community with an expanded role.
Response
Following outlined in section 2.2 are supporting evidence that the population has outstanding health needs or is at risk from health concerns that the pharmacist could address in the community with an expanded role.
Line 180-188: Pharmacists are not commonly involved in providing pharmaceutical care services. Patients with chronic diseases often visit the pharmacies for their prescription re-fills; however, their disease conditions are not followed up or monitored by community pharmacist, in a similar manner to many other developing countries [58]. Adverse drug reaction (ADR) reporting by pharmacists is now well established in many countries [59] however, pharmacists are not involved in ADR reporting in Sri Lanka despite their high prevalence [60]. Inappropriate labelling and insufficient information for patients during the dispensing of medications at community pharmacies [57] has been an ongoing challenge for many years.
Line 196-199: A recent study conducted in Sri Lanka revealed that antibiotics can be purchased without prescriptions in different districts of Sri Lanka [64]. Moreover, this study found that the presence of a qualified pharmacist was associated with reduced non-prescription dispensing practices of antibiotics by Sri Lankan community pharmacies.
Comment 3:
In section 2.1 is there evidence that staff lack of knowledge regarding medicines is contributing to error that the pharmacist could address?
Response
Yes. The reference number 35 used in section 2.1
35. Shanika LGT, Jayamanne S, Wijekoon CN, Coombes J, Perera D, Mohamed F, Coombes I, De Silva HA, Dawson AH. Ward-based clinical pharmacists and hospital readmission: a non-randomized controlled trial in Sri Lanka. Bull World Health Organ 2018,96(3),155-164.
This study by Shanika et al., 2018 showed that the ward-based clinical pharmacy service improved appropriate prescribing, reduced drug-related problems and readmissions for patients with noncommunicable diseases.
Comment 4:
There were instances where references cited did not appear to directly support statements made in the text where they were used or were outdated and the authors should please update or cite the original studies. Examples include references 22 (line 87), 23 (90), 31 (95), 53 (170)
There are also additional systematic review studies of the pharmacists role that may be cited. E.g.
https://www.ncbi.nlm.nih.gov/pubmed/29723934
https://www.ncbi.nlm.nih.gov/pubmed/26149372
https://www.cochranelibrary.com/cdsr/doi/10.1002/14651858.CD013102/full
Response
Thank you for your suggestion. We have now revised the above suggestions in the revised manuscript. This is presented in the track change version of the manuscript.
Comment 5:
It is not clear where in the text figure 1 is cited - please can the authors make this clearer and also clarify what ‘General Practice’ means.
Response
We have now revised the above suggestions and included a description about the figure in the revised manuscript (line 218-220).
General practice pharmacists perform a range of clinical and administrative duties related to their expertise in medication use and safety; the clinical activities typically include providing drug information to practice staff, educating patients and reviewing medication. Further explanation about general practice pharmacists can be found in the following article: https://www1.racgp.org.au/ajgp/2018/august/what-can-pharmacists-do-in-general-practice
Comment 6:
The educational section is sparsely referenced in comparison to earlier sections and needs greater support with literature when advocating educational teaching methods and new PG courses - what is the impact on pharmacist learning outcomes, competency and/or care? What do regulators in other countries recommend? The Am J Pharm Education is a good source of literature in this area.
Response
We have now revised the above suggestions in the revised manuscript and included references from Am J Pharm Education in section 3.
Comment 7:
In the educational section, can the authors justify why they mention only M.Pharm or PHD when other qualifications are available. For example in the UK pharmacist can undertake prescribing qualifications.
Response
As we discussed in the manuscript pharmacy undergraduate programmes were introduced relatively recently in Sri Lankan universities and there are no postgraduate degree programmes leading to M. Pharm or PhD qualifications. These postgraduate degrees are important and relevant to Sri Lanka for those who wish to specialise in particular field of pharmacy.
Comment 8:
Please use the word ‘medication’ consistently in the text in place of ‘drug’ to improve clarity.
Response
We have now revised the above suggestions in the revised manuscript.
Comment 9:
Furthermore minor comments and edits are below
Line 14 - what does significant mean? Clarify in the text please
Line 21 - include the pharmacist
285-6: last sentence not needed?
122- reference 36 please provide error rate
126- patients who lacked their medication record or possessed an incomplete record
Response
Thank you for your suggestion. We have now revised the above suggestions in the revised manuscript. This is presented in the track change version of the manuscript.

Reviewer 2 Report
Thank you for the opportunity to review this perspective paper. It is well written and referenced. The authors reference several papers that are used to support their opinions. It may be beneficial to provide more information for some of these high impact references. For example, figures or statistics that support the conclusions you are drawing. This may make this more impactful to a reader. It may also reduce jumping from one reference to the next.
Specifically, I felt some more details could be helpful for:
- Line 74 - consider providing more detail on what criteria are used to determine efficiency / proficiency, or put another way, what are pharmacists needing to be proficient or efficient at?
- Line 90 - elaborate on barriers presented in reference 23 for why pharmacists are not utilized more in developing countries
- Line 100 - Please explain what FIP is as all readers may not be familiar, or at a minimum define the organization name before using their abbreviated name.
- Line 119 - more detail on the findings of this study may be useful to include
- Line 121 - consider rewording
Figure 1 - Where was this introduced in the text?
Author Response
REVIEWER 2:
Thank you for the opportunity to review this perspective paper. It is well written and referenced. The authors reference several papers that are used to support their opinions. It may be beneficial to provide more information for some of these high impact references. For example, figures or statistics that support the conclusions you are drawing. This may make this more impactful to a reader. It may also reduce jumping from one reference to the next.
Comment 1:
Specifically, I felt some more details could be helpful for:
- Line 74 - consider providing more detail on what criteria are used to determine efficiency / proficiency, or put another way, what are pharmacists needing to be proficient or efficient at?
Response
A certificate of proficiency in pharmacy and a certificate of efficiency in pharmacy are two different certificates awarded by the Ceylon Medical College to get registered as a pharmacists. These names are used to categorize the pharmacists. The certificate of proficiency in pharmacy is to practice in the hospital setting and a certificate of efficiency in pharmacy to practice in the community setting.
Comment 2:
Line 90 - elaborate on barriers presented in reference 23 for why pharmacists are not utilized more in developing countries
Response
We have explained reference 23 as following: The reasons for a lack of clinical pharmacist-led activities in hospitals in these countries are complex but may reflect the emerging nature of the pharmacy profession in these countries, professional isolation and a lack of recognition of the role of clinical pharmacy within the health care system [23].
Comment 3:
Line 100 - Please explain what FIP is as all readers may not be familiar, or at a minimum define the organization name before using their abbreviated name.
Response
We have explained this abbreviation in the revised manuscript. FIP: International Pharmaceutical Federation.
Comment 4:
Line 119 - more detail on the findings of this study may be useful to include
Response
We have now included more detail of the findings (line 121 -122).
Comment 5:
Line 121 - consider rewording
Response
We have now edited this in the revised manuscript.
Comment 6:
Figure 1 - Where was this introduced in the text?
Response
We have now included a description about the figure in the revised manuscript.

Round 2
Reviewer 1 Report
Many thanks for submitting a revised paper.
Only minor typographical changes now needed at galley proof stage, e.g.
line 124 - inappropriate documentation for prescribed...
line 217 - The diagram shows that the pharmacists'...
Please review the proof carefully
Author Response
Thank you for your suggestion. We have now revised your suggestions in the revised manuscript. This is presented in the track change version_review round 2 of the manuscript.
